# Smart Mobility: The Main Drivers for Increasing the Intelligence of Urban Mobility

**Paulo Antonio Maldonado Silveira Alonso Munhoz** [1], **Fabricio da Costa Dias** [1],
**Christine Kowal Chinelli** [1], **André Luis Azevedo Guedes** [1,2],
**João Alberto Neves dos Santos** [1], **Wainer da Silveira e Silva** [1]
**and Carlos Alberto Pereira Soares** [1,*]

[1]  Pós-Graduação em Engenharia Civil, Universidade Federal Fluminense, Niterói RJ 24210-240, Brazil;
     paulo_munhoz@id.uff.br (P.A.M.S.A.M.); fabriciodias@id.uff.br (F.d.C.D.); cchinelli@id.uff.br (C.K.C.);
     andre.guedes@unisuam.edu.br (A.L.A.G.); joaoneves@id.uff.br (J.A.N.d.S.);
     wainer_silva@id.uff.br (W.d.S.eS.)
[2]  Departamento de Ciência da Computação, Centro Universitário Augusto Motta,
     Rio de Janeiro RJ 21041-010, Brazil
*   Correspondence: capsoares@id.uff.br

**Abstract:** Urban mobility plays a key role in the ecosystems of complex smart cities. It is considered a key factor in enabling cities to become more intelligent, which highlights the importance of identifying the drivers that improve the intelligence of cities. In this study, we investigate the main drivers with the potential to increase urban mobility intelligence and assign them a priority. Following on from a systematic review of the literature, we conducted broad and detailed bibliographic research based on the recommendations of the Preferred Reporting Items for Systematic Reviews and Meta-Analysis (PRISMA). We also surveyed 181 professionals working in the field concerned to confirm the importance of different drivers and assign them a level of priority. The results show that 27 drivers identified in the literature were considered important, of which seven, related to city governance and technical solutions, were considered the most important to increase urban mobility intelligence.

**Keywords:** smart mobility; intelligent mobility; urban mobility; smart transport; intelligent transport; sustainable transport; smart city; intelligent city; drivers

---

## 1. Introduction

Smart mobility has become an increasingly present theme in sustainability agendas in response to the impacts of transportation systems in cities. The concept of smart mobility has evolved mainly from the convergence of the digital revolution with the transport industry. Thus, new technologies have been used to increase transport network efficiency [1], notably those related to information and communication technology. In the literature, the concept of smart mobility, which has also been addressed as intrinsically associated with smart cities [2–7], is considered an essential driver for increasing the intelligence of cities [8].

There has been an increase in the use of" smart city" terminology in academia; however, there is no consensus regarding the definition of the smart city concept [8–10]. We agree with Guedes et al. [8], who states that "a more current and comprehensive way of understanding a smart city from the integration of existing knowledge and experiences is that of an innovative city, which combines aspects of intelligence and sustainability through a governance that integrates stakeholder interactions and uses the technology." The" smart city" concept has great potential to address several adverse effects of

rapid urbanization [3] because it is linked to the implementation of several beneficial changes in the functioning of city dynamics [11].

The availability of information and communication technologies (ICTs) and digital technologies has contributed to the dissemination of improvements linked to the concept of the" smart city" [12]. Government agencies have begun to invest in information technology systems, such as applications for individual use, cameras in monitoring systems and sensors, aiming at positive changes in citizens' daily lives [13]. In this context, the use of ICTs in the field of urban mobility has played a prominent role in promoting more sustainable and efficient transport [14], as, for example, in the case of technologies that allow the use of interconnected services, such as carsharing, bikesharing, ridesharing, buses and trains, in real time, promoting multimodality [15,16]. The use of these technologies in combination with the need for more flexible mobility and for low $CO_2$ emissions has resulted in the dissemination of these new initiatives in the field of mobility [17], thus making such mobility more intelligent.

The new technologies used for urban mobility are usually aimed at improving sustainability, which is a factor of great relevance for smart cities. Protection of the environment is an essential aspect in the formulation of public policies, which in urban mobility manifests as a transition to a low-carbon circular economy and reductions in emissions [18]. Actions to improve urban mobility must be aligned with concepts related to sustainability. In this sense, active participation by the state and civil society is necessary to change or improve the transport network, generating positive impacts for all involved [19].

Technocentric approaches to smart cities are the most commonly used approaches in the literature; however, they can result in solutions that are prejudiced against sustainability goals [1,7]. Therefore, as they involve complex issues, the approach to smart cities should, in addition to analyzing solutions from a technological perspective, utilize smart solutions that take into account aspects of sustainability and analyze the reality of each city, considering the geographical conditions, ecosystems and availability of resources in the region, so that the implementation of solutions can be undertaken confidently [9].

Although the concept of urban mobility is not new, we did not find articles in the literature that aimed to identify drivers that contribute to increasing mobility intelligence considered as a whole. We did find some studies addressing issues related to smart mobility. We were able to identify factors that could contribute to increasing mobility intelligence in the discussions and results of several of these studies.

In this context, we address two main issues in this paper. The first is: what are the main drivers for increasing urban mobility intelligence? To address this question, we performed broad and detailed bibliographic research considering the recommendations of the Preferred Reporting Items for Systematic Reviews and Meta-Analysis (PRISMA). The second issue is: what are the priority drivers for increasing urban mobility intelligence? To addresses this question, we surveyed 181 professionals working in the concerned field in order to confirm the importance and priority of drivers.

The major importance of identifying the drivers that make the mobility of a city smarter is that this makes it possible to analyze and establish a relationship indicating the priorities of drivers such that actions aimed at the improvement of mobility can be undertaken together with the bodies responsible for urban transport management. Therefore, by using analyses like this, the public agencies responsible for mobility can acquire an understanding of which factors are most important in their contributions to the mobility of a smart city and thus carry out investments and initiate improvement projects for the infrastructure of cities with greater knowledge and greater confidence. The results of this work can also be used by transportation companies and technology startups to present models and possible partnerships with the government that put in place solutions for mobility.

This paper is structured as follows: Section 2 presents a review of the literature, mainly addressing the drivers identified in the bibliographic research. Section 3 presents the procedures used to carry out the bibliographic research, identification of smart mobility drivers, survey of expert opinions and the data analysis. Section 3 presents and discusses the research results. Conclusions are provided in Section 5.

## 2. Literature Review

The rapid growth of urban populations together with high demands on quality of life has created a need for improvements in all spheres of infrastructure and subsystems within cities [20]. The increased demand for mobility from citizens leads to problems such as congestion and pollution [19,21]. However, urban mobility has the potential to move from a critical problem to a possible source of improvement and transformation for modern cities [22]. For the transformation to be well structured, the city needs to have a long-term vision that is usually represented by an Urban Mobility Plan to ensure smart solutions come to fruition [4]. The planning of urban mobility in the contemporary world must begin considering the need to promote safer, integrated, and, above all, sustainable, means of transport [23].

Transport planning has evolved considerably with the growth of large cities and the dynamics of land occupation, resulting in new mobility solutions that allow access to all regions of the city [24]. Intelligent mobility solutions can offer users more transport options and more adaptable and affordable travel while reducing the reliance on private vehicles and promoting energy-efficient mobility [7,25]. Taking into account that transport is responsible for 20% of energy consumption in a city, investments in intelligent transport systems show promising results in terms of gas reduction and energy savings [4] and, for this reason, several studies in the field of smart mobility are related to sustainable thinking [1,7,14].

To transition from driving cities to smart transport, it is up to the government to be proactive and invest in sustainable transport solutions that promote energy saving [4]. However, as with any socio-technical transition, critical questions must be asked in terms of how the transition to smart mobility will be managed and how the benefits and any negative externalities of change will be governed [26]. In this complex scenario, the management of mobility by public and private agencies requires the adoption of intelligent and dynamic decision-making approaches capable of controlling various issues that change in real time [26], such as barriers related to the administrative regime of cities and governance incompatible with or inadequate for the new initiatives [19,27]. Decision-making by responsible bodies must be linked to information such as the amount of data available, intelligent services, and energy results. They must also stick to issues that are specific to the reality of each city to implement relevant solutions that make mobility more intelligent [28].

Thus, it is very difficult for researchers to establish a unique concept for intelligent mobility [7,23]. Although the concept of smart mobility is increasingly addressed, there is still no consensus among researchers, as shown in Table 1.

**Table 1.** Definitions of smart mobility by researchers.

| Definitions of Smart Mobility | Authors |
|---|---|
| It is the area of a smart city representing broadly defined mobility, the components that comprise the traditional understanding of the transport of people and goods, and the dissemination of information by digital means. | [11] |
| Intelligent mobility is a comprehensive concept that makes the transport network's sustainability more achievable due to the search for improvements in transport services, balancing the application of technology with social, economic, and environmental aspects. | [18,28] |
| It is the ability to access transportation services from integrated platforms that aggregate the community and present intense processing of data from users to match the demand forecast. The infrastructure must be smart with connected and sustainable vehicles. | [19] |
| It is the integration of sustainable, intelligent, and cooperative vehicle technologies with a cloud server and vehicle networks based on big data. | [29] |
| A sustainable and safe environment that meets the mobility needs of citizens and integration with intelligent systems to provide traffic information. | [30] |
| Smart mobility is related to transport and use of communication and information technologies to promote accessibility and increase the quality of life. | [31] |
| It is the application of solutions that combine behavioral economics, e-participation, and crowdsourcing to obtain better energy consumption when moving around the city. | [32] |
| It is the adoption of digital technologies that make mobility services in a city or territory more accessible and easier for citizens. | [33] |

From these concepts, we can understand smart mobility as being mobility that uses digital technologies to integrate systems and means of transport that interacts with users, aiming at a sustainable, safe, accessible environment that meets citizens' mobility needs.

Through bibliographic research, we identified the following drivers to have the potential to increase mobility intelligence: urban mobility plans, public policies, city expansion dynamics, multimodal integration, ride and vehicle sharing: use of the bicycle, scooter, and carsharing systems, alternative transportation, information and communication technology (ICT), data collection, smart traffic lights, smart parking, flexible pass payment, cybersecurity, technological innovation, integration of the city's intelligent infrastructure with smart buildings, public accessibility to real-time information, open data, environmentally-friendly policies, accessibility, walkability, selective ranges, safety, logistic solutions, cooperation between stakeholders, continuous renovation of transport infrastructure, maintenance, traffic accident detection and support system. In the next paragraphs, we will give a summary of each one.

The urban mobility plan is a public management instrument that guides short-, medium-, and long-term projects, investments, and actions. It translates the objectives of improving local urban mobility into goals and strategic actions [23]. Intelligent mobility requires a participatory process for preparing the plan, in which stakeholders identify challenges and propose improvement actions. They must also correlate the evolution in the usage dynamics of urban spaces with the transport system and consider the effect of new interventions on the natural and built environment and citizens' lives [7,23]. Intelligent Urban Mobility Plans incorporate the intensive use of technology to improve the efficiency and effectiveness of the system and interaction with the user.

Public policies establish guidelines for the state's actions, decisions, and programs to fully develop the city's functions and the encouragement of smarter urban mobility solutions. It is essential that their guidelines enhance projects to expand the mobility system and incorporate new technologies that enable more intelligent, sustainable and accessible solutions [1,7]. In this sense, they must create a favorable environment for investments, regulating the financing of the system and the legal and financial guarantees. Projects to improve the system's intelligence, enhanced by public policies, should also improve the quality of life and access of citizens to transport services [12].

City expansion dynamics are related to the transformations in cities' structure and shape mainly based on population growth and land use and occupation [25]. Urban growth and mobility are intrinsically related factors, enhancing each other. The influence of mobility on the pattern of land use [3] means that planning for smarter mobility should consider this aspect, especially concerning more intelligent solutions for the transport network design [25].

Multimodal integration aims to optimize the transport of people and goods [13]. It is a key factor for smart mobility and is associated with the increasing replacement of vehicle 'ownership' by 'usership,' the interconnection of all mobility services and the improvement of sustainability [15]. It is also associated with equity in citizens' access to public transport [34].

Ride and vehicle sharing are related to the use of bicycles, scooters, and car-sharing systems. The growing expansion of sharing systems worldwide and their adaptability are attractive indicators for citizens' greater use [35]. Sharing systems can be used more and more from the moment that safe road infrastructure has been developed [36].

Alternative transportation focuses on reducing car dependency. In this sense, unconventional forms of transport are also used to improve traffic and sustainability. The main benefits are the reduction of traffic jams and expenditure on energy and the reduction of costs and time spent traveling for citizens [4]. However, this type of solution requires planning that integrates the different alternatives and ensures user safety [37].

ICTs are crucial for successfully implementing smart city mobility [13], as they are the main instrument for remote control and management of the mobility system. They integrate services, equipment, and applications to transmit, store, create, share or exchange information, giving a central structure of control and command to governance, supporting transport operators and users' decisions [7].

They are also important instruments for improving vehicles and people's movement, mainly due to the services optimization, increased user interaction with the traffic system, and integration of intelligence and sustainability aspects [13,14].

Data collection refers to the process of collecting real-time data related to mobility. Smart services and the transport network used by citizens, with their movements and perceptions, provide data for managers to improve mobility services [22]. ICTs play an essential role in expanding the possibilities for collecting and monitoring data in real-time [7], which can be used to monitor and manage the mobility infrastructure [6]. Databases are fundamental for increasing the efficiency of services, such as traffic analysis, route mapping, demand analysis, and ticketing [38]. They also improve the user's interaction with the transit system and integrate intelligence and sustainability [13,14].

Smart traffic lights are related to the traffic control system that combines traditional traffic lights with sensors and artificial intelligence to optimize the operation of traffic lights depending on the flow and number of vehicles on the roads [9,32]. In the context of smart cities, traffic lights that can adapt to the presence or absence of vehicles are essential in the daily commuting of citizens to and from work [39]. The implementation of these solutions reduces travel time and increases the sustainability of mobility, mainly due to the reduction of emissions [40].

Smart parking is a parking strategy that integrates the use of road sensors, intelligent displays, cameras, parking meters, and software to inform about the parking spaces' availability and manage their use [14]. The concept of intelligent transport systems (ITS) is related to smart parking, considering technological innovation and sustainability in transport [14]. The technologies incorporated into smart parking lots optimize the search for parking spaces, reduce congestion, enable electronic verification of parking permits [20], and facilitate parking payment through electronic devices [23].

Flexible pass payment is related to flexible ticket payment systems that enable payment by applications and do not necessarily require physical money [38]. In the context of smart mobility, users have access to their mobility information and efficient payment systems electronically or online [38]. Payment is made transparently for transport services on simple or multimodal trips, parking spaces, tolls and use of public bicycles [41]. Flexible payment can also be related to sustainability when the payment system analyzes the performance of the vehicle and establishes rates according to the emissions of gases benefiting the most sustainable vehicles [42].

Cybersecurity is the protection of mobility systems' management and operation components, such as computers, networks, programs, and data, against digital attacks. It includes unauthorized access, data alteration or destruction, and services interruption or disorientation. The increase in intelligence implies high data dissemination that needs technical protection [38]. Issues such as privacy of stored data, real-time location, daily citizen routes, contact lists, and messages are the primary problems associated with cybersecurity [43]. The lack of knowledge about the technology and the concern with safety in using the systems make it challenging to improve intelligent solutions [44]. The term "security" in the context of cybersecurity is often defined as protecting the integrity, confidentiality, and accessibility of information [45].

Technological innovation is at the center of discussions about increasing smart city ecosystems' intelligence, in which mobility plays a key role. They represent the development or improvement of new technologies and their implementation in processes and products that collaborate to increase mobility intelligence [10]. A smart city aims to increase the quality of life of its citizens through smart technologies [9,10]. In this context, it is the incorporation of newly innovated technologies that improve mobility systems' capacity to enhance this objective.

The integration of the city's intelligent infrastructure with smart buildings is a key factor for smarter ecosystems. In this context, integrating the building's smart systems with the city's intelligent systems provides a more intelligent urban system [17]. Buildings in the context of smart mobility are designed to take full advantage of the potential of the transport network and to accommodate product deliveries and returns [25] intelligently. Building users can use the city's smart grids to, for example, get information about traffic and availability of parking spaces. On the other hand, intelligent mobility

systems can benefit from information about building occupancy and times with the highest entrance and exit flow

Public access to information in real-time is crucial for the user to interact with transport systems. With the data collected from these systems, it is possible to provide real-time information, such as hourly tables and forecasts at public transport stops, and road conditions. It makes it possible for citizens to decide on the most suitable routes and means of transport, making trips more adapted to user needs [14].

Open data is an essential aspect of information management and the starting point for the emergence of innovative solutions based on information technology to improve city services [10,28]. In this context, government officials must have data opening policies that guarantee the availability of information from mobility systems to anyone who wants to access it, allowing, for example, service providers to use this data to create mobility alternatives and that users have access to system performance. Open data requires high-density networks [46] and the semantic integration between opening data, private data and real-time data from different city service operators [28].

Environmentally-friendly policies are organizational practices based on laws, regulations, guidelines, and other policy instruments related to environmental issues to reduce the transport impacts on ecosystems [10]. They are usually associated with the reduction of greenhouse gas emissions, air pollution, and noise. Intelligent locomotion depends on an efficient means of public transport with the least possible impact on the environment [23]. In a smart city scenario with a broad approach to digital technologies, initiatives are essential for companies to include environmental goals in their mobility projects [47].

Accessibility is related to guaranteeing safe and autonomous access for people with disabilities or reduced capacity to the transport infrastructure [21]. In the context of mobility, smart applications increase citizen autonomy by improving the process of detecting barriers to displacement and identifying specific facilities for people with special needs [21]. In smart mobility, the improvement of accessibility occurs mainly due to better urban projects, the incorporation of new technologies, support policies, economic incentives, and city leaders' involvement [7].

Walkability is related to the incentive that citizens receive to use walking as a means of locomotion. It is influenced mainly by the fluidity of the walk provided by the roads' conditions and the possibilities of integration with other means of locomotion. In parallel with introducing new urban mobility technologies, infrastructure projects for new areas, and urban reforms must follow the patterns of transitable, safe, accessible and pleasant means of transport [48]. Intelligent mobility presupposes more pedestrian-friendly cities [49] since walking is the most basic and environmentally friendly means of transport [48].

The selective ranges are traffic lanes with traffic restrictions for some means of transport, aiming at the flow of traffic on the roads [27]. They are used for certain types of mode and can stimulate alternative transportation systems and vehicle sharing, with the option of using the lanes at certain times [50].

Safety is related to the actions aimed at improving the security of the transport infrastructure to ensure the physical integrity of the users of the roads and means of transport, as well as preventive actions. For citizens, security is one the most important elements in smart cities [20]

Logistic solutions optimize the commodity logistics chain, traffic networks, freight, and goods movement [46]. Urban logistics in smart cities focus on implementing more economical routes, both financially and through reducing emissions [6]. Digital innovation has been an important tool for adding intelligence to logistics solutions, increasing the focus on citizens' experience and expectations about mobility and the possibilities of routes offered by georeferenced applications [51].

Cooperation between stakeholders refers to the inclusion of society's segments and public and private bodies that may affect or are affected by mobility services' results in discussions on improving these services to ensure that the innovation's implementation and mobility-related interventions meet all interests. The insertion of new mobility technologies is strongly influenced by technological

innovation in which innovation experiments take place in the transition from real life and are driven by social challenges and the stakeholders' needs and expectations [17]. They can be involved in various transportation planning stages, such as improving bus routes and schedules, identifying unnecessary bus stops, or adding new ones [52]. The process of involving stakeholders with different profiles and understandings creates a scenario in which mobility needs must be aligned [26].

Continuous renovation of transport infrastructure concerns the maintenance and modernization of the current infrastructure and the implementation of new solutions to ensure the capacity to meet travel demand. The transport network structural conditions influence citizens' daily lives [4] and the opportunities for incorporating innovative technologies that enable more intelligent solutions, such as those related to new materials and ICTs. On the other hand, the smarter the mobility, the greater its capacity to adapt to innovative technologies [19].

Maintenance includes actions to conserve roads, vehicles, equipment and systems used in the mobility of cities [53]. It is strongly influenced by the standards of construction and equipment used on public roads, as well as by the intensity and typology of the vehicle flow [54]. Intelligent mobility focuses mainly on predictive maintenance which, through collecting data from scattered sensors and cameras, supplies maintenance systems with data on the operating conditions of roads and equipment [53].

Traffic accident detection and support systems integrate hardware, software, procedures, and facilities for real-time detection of traffic accidents and rapid assistance to the injured [4]. New technologies, such as cameras with better long-range night vision, sensors, and drones, can increase the system's ability to identify the context and severity of accidents, helping to reduce their impact on the victim's life and traffic conditions [55,56].

## 3. Materials and Methods

We use an approach widely used in research that aims to identify variables related to a given phenomenon and the degree of importance of these variables. It consists of five steps: bibliographic research, identification of smart mobility drivers, survey of expert opinions, data analysis, and structuring the presentation of results and discussion. Figure 1 summarizes the five-step methodology.

### 3.1. Bibliographic Research

We conducted broad and detailed bibliographic research on the Web of Science, Scopus, Scielo, and the main scientific journals' websites, covering works published in the last 10 years, using the keywords smart mobility and intelligent mobility.

We took into account the recommendations of Webster and Watson [57], Guedes et al. [8], and the Preferred Reporting Items for Systematic Reviews and Meta-Analysis (PRISMA) [58], which aims to improve the outcome of systematic reviews and meta-analyzes.

Initially, we performed an exploratory reading considering the most relevant titles, abstracts, and keywords to select the studies that were adherent to the theme of this work. As a result, of the 834 articles identified, we discarded 590 articles whose abstracts were not relevant to the topic researched, were not peer-reviewed, or were not available in their entirety for reading. Then, we carried out selective reading in the remaining 244 articles to check if this perception of the abstracts was correct and if they were articles relevant to the topic. As a result, we excluded 76 articles that were not original, whose results did not contribute to the theme, and whose results were not supported by the methodology.

Finally, we carried out a detailed reading of the 168 remaining articles, of which 141 articles were used to support this work. Figure 2 summarizes the bibliographic search from the PRISMA diagram.

Although all selected articles contributed to substantiate this work, some contributed more than others. Below we will present a summary of each one, consolidating excerpts written by the authors.

**Bibliographic Research**

Screening article´s title and abstract using keywords

Quick reading of selected articles to confirm your relevance

Detailed reading of works considered important

Works published in the last 10 years
- Web of Science
- Scopus
- Scielo
- The main scientific journals' websites

**Identification of smart mobility drivers**

Screening article´s title and abstract using keywords

Quick reading of selected articles to confirm your relevance

Detailed reading of works considered important

- Critical and reflective reading of the articles
- Potential driver information in spreadsheet cells: authors in rows and drivers in columns
- Potential drivers: portrayed in at least one more work without reference to each other

**Survey of Expert Opinions**

Questionnaire
- Structured in an online platform
- Five-point Likert scale (extremely important to minimally important)
- Questions about demographic data and addressing the drivers importance

**Pre-test**: validation of the questionnaire's overall design and the questions' clarity and pertinence

- To have at least five years of activities related to transport and urban mobility fields
- To have at least an academic master's degree;
- To hold a degree in one of the following College of Knowledge: Humanities Sciences, Life Sciences, Exact, technological and multidisciplinary sciences.

**Data Analysis**

Reliability assessment

Prioritization of drivers

- Cronbach's Alpha: analyses of variance attributed to the respondents and the variance attributed to the interaction between respondents and items
- Prioritization using relative median
- Illustrations summarizing the results for improving analysis

**Results and Discussions**

Use of Graphs and tables to present the results

Discussion the results focusing on the priority drivers in the context of the Brazilian reality

- Respondents` profile
- Drivers classified by the relative median
- Drivers ranked as "extremely important
- Drivers' behavior

**Figure 1.** Synthesis of the methodology.

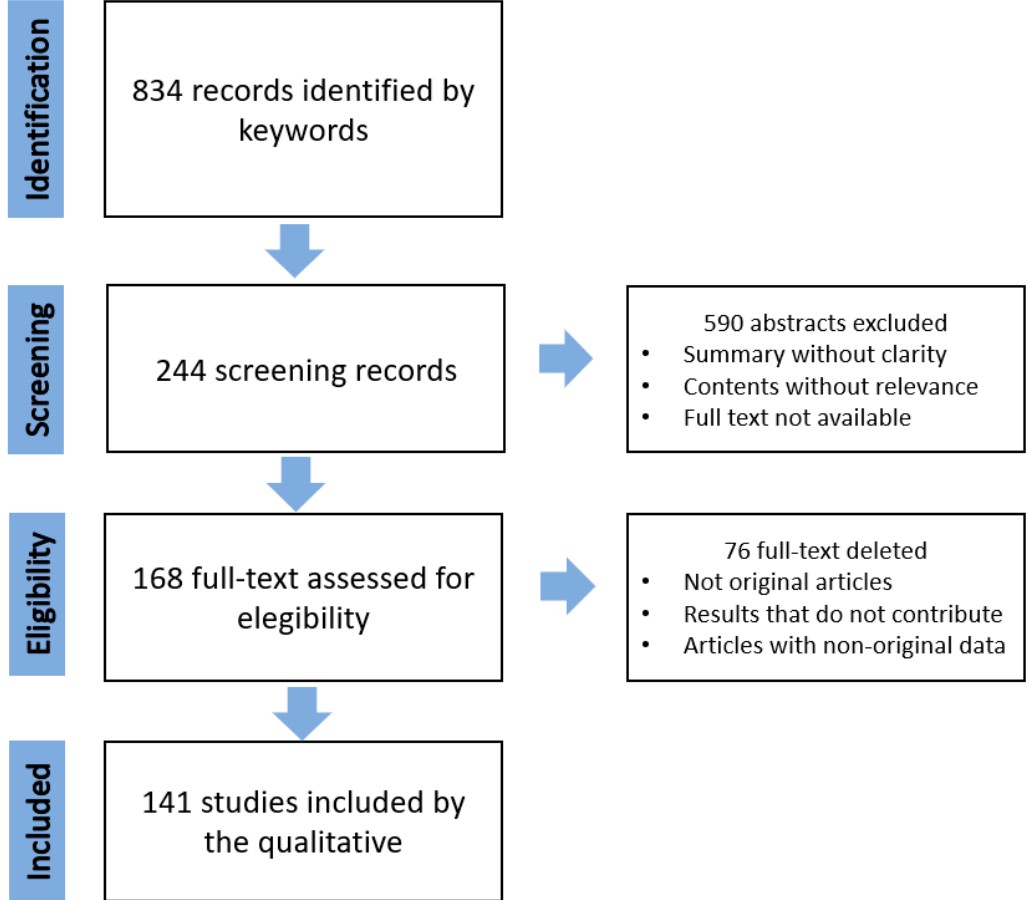

**Figure 2.** Literature search from the Preferred Reporting Items for Systematic Reviews and Meta Analysis (PRISMA) flowchart.

Getting smart about urban mobility—aligning the paradigms of smart and sustainable [7]: the paper puts forward and explores the definition of smart urban mobility to help draw the paradigms of smart and sustainable closer together towards a common framework for urban mobility development. This article seeks to get to the bottom of what we think we mean or should really mean when we label urban mobility 'smart.' Its writing has been motivated by a concern that we may be at risk of reserving smart for a focus on technology enablement. There is a need to ensure that the paradigms of smart urban mobility and sustainable urban mobility are aligned.

Smart Cities Concept: Smart Mobility Indicator [11]: the authors analyze and discuss the solutions employed as smart mobility solutions to test smart cities' effectiveness. As a result of the analysis, the author proposes an indicator to analyze which solutions are more intelligent within those possible to implement, to improve urban mobility. The indicator is based on parameters in the following areas: transport infrastructure, information infrastructure, mobility methods and vehicles used, and legislation.

A comprehensive view of intelligent transport systems for urban smart mobility [14]: the authors conduct an in-depth analysis of intelligent transport systems' role in supporting urban mobility to identify the main knowledge gaps in the literature. Most of the articles analyzed were related to technology and, according to the author, there was a lack of quantitative models of analysis.

Smart mobility transition a socio-technical analysis in the city of Curitiba [27]: the authors studied the case of Curitiba, located in the south of Brazil and recognized as a smart city, stands out for its pioneering project of the Bus Rapid Transit (BRT) system. They analyze urban mobility based on the socio-technical transition of innovation and the multi-level perspective. They also have developed

a timeline of the initiatives considered as milestones towards the transition to smart mobility and proposed a reflection about the socio-technical innovation approach applied to smart urban mobility.

Encouraging sustainable mobility behaviour by designing and implementing policies with citizen involvement [32]: the paper discusses the theoretical concepts, design considerations and preliminary findings from smart mobility. It also presents the results of a project that aims to devise measures to encourage the increased use of public and non-motorized transport, integrating the principles of behavioral economics in public policies. The article describes the energy policies behind the project and outlines the theoretical framework for integrating behavioral perceptions in technology design and public policy.

An Interdisciplinary Review of Smart Vehicular Traffic and Its Applications and Challenges [39]: the authors discuss the main application areas of smart traffic and smart mobility, synthesizing different perspectives. Some research challenges pertinent to sustainability, insurance, simulation and the handling of ambiguous information are also highlighted.

A taxonomy for planning and designing smart mobility services [41]: the authors studied the state of the art in mobility services through publications on intelligent mobility services provided by cities worldwide. The author proposed a taxonomy to plan and design intelligent mobility services considering the following dimensions: type of services, maturity level, users, applied technologies, delivery channels, benefits, beneficiaries, and common functionality.

### 3.2. Identification of Smart Mobility Drivers

To identify potential intelligent mobility drivers, we performed a critical and reflective reading of the articles selected in the bibliographic research. When we found an article with information about a potential driver, we extracted this information into the spreadsheet cells. The rows contained the analyzed articles and the columns the drivers. To increase the chances that a driver was really important, we adopted the strategy that its importance should be portrayed in at least one more work and that these works did not reference each other. All drivers overcame this condition since five studies referenced the drivers with the least reference. As a result, we identified 26 drivers subjected to expert assessment to determine each driver's importance degree for increasing mobility intelligence.

### 3.3. Survey of Expert Opinions

We used an online platform (Google Forms) to structure a questionnaire containing three sections: (a) conceptualization of the drivers; (b) questions regarding demographic data; (c) questions addressing the importance of drivers identified in the literature research. The experts expressed their opinions on each driver's degree of importance to make the mobility smarter, using a five-point Likert scale, ranging from extremely important to minimally important. The drivers were presented at random to avoid responses being influenced by the order in which they appeared.

We invited professionals who work in the concerned field to answer the pre-test and the reviewed questionnaire. The inclusion criteria were: (a) to have at least five years of activities related to transport and urban mobility fields, since, in Brazil, this is the minimum time of professional experience necessary to perform specific activities that require more specialized knowledge; (b) to have at least an academic master's degree; and (c) to hold a degree in one of the following "College of Knowledge": Humanities, Life Sciences, or Exact, Technological and Multidisciplinary Sciences.

The College of Knowledge is a classification by the Coordination for the Improvement of Higher Education Personnel (CAPES), Brazil, which group the training areas according to the affinity of their objects, cognitive methods, and instrumental resources. Humanities covers areas such as architecture and urbanism, administration, economics, urban and regional demography, political science, education, and geography. Life Sciences covers areas such as biological sciences, medicine, and public health. Exact, Technological and Multidisciplinary Sciences covers areas such as engineering, environmental sciences, geosciences, statistics, and chemistry.

The pre-testing of the questionnaire survey was performed from 11 May to 13 July 2020, with seven experts who expressed their opinion about the questionnaire's overall design and the questions' clarity and pertinence.

To identify the specialists who could participate in the research, we adopted two strategies. The first was to search the masters and doctorate courses in transport and urban mobility for the teachers' curriculum to verify their experience in the fields of transport and urban mobility. Through the teachers' curriculum it was also possible to identify the co-authors of scientific productions and expand the potential respondents' list. As a result, we identified 400 experts. The second strategy was to access the leading Brazilian journals that deal with transport and urban mobility to identify authors who work with the theme. As a result, we identified 200 experts. Thus, 600 professionals were invited by email, of which 181 professionals participated in the survey from 14 July to 25 October 2020.

*3.4. Data Analysis*

To evaluate the questionnaire's reliability and the respondents, we used Cronbach's Alpha [59]. To prioritize the drivers, we use the concept of the relative median [8], using the formula below:

$$Rm = \{1 + \frac{Pr}{j_1} \; for \; m = 1 \quad m + \frac{Pr - \left(\sum_{i=1}^{m-1} j_i + 1\right)}{j_m} \; for \; 2 \le m < N \; and \; m =$$
$$integer \quad m + 0.5 \; for \; 1 \le m < N \; and \; m = Fractional \; number \quad N \; for \; m = N\}$$

where: $Rm$ is the relative median, $m$ is the median, $Pr$ is the relative position of the median, $j_i$ is the number of respondents assigned a semantic classification of "$i$," and $n$ is the maximum value of the Likert scale used.

The concept of the relative median considers the distance from the median to the nearest class. As an example, in Figure 3, the median value of four in the first line is much closer to the frequency represented by the number three. In the second row, the median is shifted to the right as we add more cells to the frequency defined by the number five. Although the two lines have medians equal to four, the second line driver can be interpreted as more important since it received more ratings like five and maintained the other frequencies.

| 1 | 2 | 2 | 3 | 3 | 3 | 3 | 3 | 3 | **4** | 4 | 4 | 4 | 4 | 4 | 5 | 5 | 5 | 5 | | | | | | | | | | |
| 1 | 2 | 2 | 3 | 3 | 3 | 3 | 3 | 3 | 4 | 4 | 4 | 4 | 4 | **4** | 5 | 5 | 5 | 5 | 5 | 5 | 5 | 5 | 5 | 5 | 5 | 5 | 5 | 5 |

**Figure 3.** Example of the median position.

The results were grouped according to the three Colleges of Knowledge. For a driver to be considered important for increasing mobility intelligence, its relative median should be greater than 3.0. To be considered a priority, your relative median should be equal to 5.0 for the three Colleges of Knowledge and the entire sample.

*3.5. Structuring of the Presentation of Results and Discussion*

We use the strategy of using tables and graphs to synthesize the information as much as possible. The flow used was: (a) to present a table with the drivers, a summary of their conceptualization and the works that referenced them; (b) show the respondent's profile; (c) use graphs and tables to present the experts' answers on the degree of importance of each driver, aiming to validate the view of researchers who work with the theme, identify the priority drivers, based on the criteria set out in Section 3.4. Data analysis, and the drivers' relative behavior for the three Colleges of Knowledge and the entire sample; (d) group priority drivers according to their approach; (e) discuss the results focusing on the priority drivers in the context of Brazilian reality.

## 4. Results and Discussions

### 4.1. Selected Drivers

Twenty-six drivers were selected according to the criteria established in materials and methods. Table 2 shows the definition of the drivers found and the articles that referenced them.

**Table 2.** Selected drivers.

| Drivers | Sources |
|---|---|
| I. Urban Mobility Plans: Is public management instrument that guides short, medium, and long term projects, investments, and actions. Translates the objectives of improving local urban mobility into goals, strategic actions. | [4,10,18,23,27,34,60–70]. |
| II. Public Policies: Establish guidelines for the State's actions and decisions and programs to fully development of the city's functions and the encouragement of smarter urban mobility projects. | [1,7,12,19,21,22,24,26,27,29,37,41,42,47, 60,63–93]. |
| III. City Expansion Dynamics: Dynamics of transformations in cities' structure and shape based on population growth and land use and occupation. | [3,25,62,63,77,86,87,92,94,95]. |
| IV. Multimodal Integration: Integration of transport modes to optimize the transport of people or goods. | [5,13,15,34,37,41,52,63,64,66,75,76,86, 87,96–103]. |
| V. Ride and Vehicle Sharing: Use of the bicycle, scooter, and carsharing systems. | [2,4,5,11,15–17,21,23–25,33,35–37,41, 43,45,61,64,67,71,77,79,80,85,89,98,99, 103–109]. |
| VI. Alternative Transportation: Use unconventional forms of transportation to reduce car dependency and improve traffic and sustainability. | [1,4,11,23,33,36,37,71,80,90,93,102–104, 110–114]. |
| VII. Information and Communication Technology: Integrates services, equipment, and applications to transmit, store, create, share, or exchange information, giving a central structure of control and command to governance, supporting transport operators and users' decisions. | [1,3,6,7,9,11–17,22,29–32,35,38,43,54, 55,60,63,64,66,70,85,86,92,102,104,115–118]. |
| VIII. Data Collection: Process of collecting real-time data related to mobility to be used to monitor and manage the mobility infrastructure, and increase the efficiency of services, such as traffic analysis, route mapping, demand analysis, and ticketing, and improve the user's interaction with the transit system. | [4,10,14,33,39,46,53,72,81,92,116,119–125]. |
| IX. Smart Traffic Lights: The traffic control system that combines traditional traffic lights with sensors and artificial intelligence to optimize traffic signs' operation depending on the flow and number of vehicles on the roads. | [9,14,31,39,41,43,71,107,126]. |
| X. Smart Parking: Parking strategy that integrates the uses of sensors, cameras, parking meters, and software to inform about the parking spaces' availability and manage their use. | [14,20,68,84,107,118,125–129]. |
| XI. Flexible Pass Payment: Flexible ticket payment systems that enable payment by applications and do not necessarily require physical money. | [26,38,40–42,52,61,85,97,98,102,107, 111]. |
| XII. Cybersecurity: Protection of mobility systems management and operation components, such as computers, networks, programs, and data against digital attacks. Includes unauthorized access, data alteration or destruction, and services interruption or disorientation | [38,40,42–45,47,54,83,100,101,106,119, 129]. |
| XIII. Technological Innovation: Development or improvement of new technologies and their implementation in processes and products. | [1–4,7,9,10,13,19,23,25,28–34,37,39,42, 43,45,48,50,51,62,70,73,74,81,85,90,91, 93,99–101,118,121,124–126,130]. |
| XIV. Integration of the city's intelligent infrastructure with smart buildings: Integration of building's smart systems with the city's intelligent systems for more intelligent urban systems. | [25,96,131–133]. |
| XV. Public accessibility to real-time information: Ensuring population access to information in real-time, such as hourly tables and forecasts at public transport stops and road conditions. | [2,4,9,14,20,28,41,56,64,72,85,94–98, 107,118,124]. |
| XVI. Open Data: Data opening policies that guarantee the availability of information from mobility systems to anyone who wants to access it. | [5,10,12,28,38,43,46,62,92,120,127,134]. |
| XVII. Environmentally-friendly policies: Organizational practices based on laws, regulations, guidelines, and other policy instruments related to environmental issues to reduce the transport impacts on ecosystems | [1,2,10,17–19,23,26,27,29,30,32,47,50, 52,60–64,67,70,74,78,83,93,98,110,114, 117,135,136]. |

**Table 2.** *Cont.*

| Drivers | Sources |
|---|---|
| XVIII. Accessibility: Guarantee of safe and autonomous access for people with disabilities or reduced mobility to the transport infrastructure. | [1,2,7,21,27,48,64,88,96,109,117,132, 137]. |
| XIX. Walkability: Encouraging walking as a means of travel by improving the fluidity of walking and safe conditions for sidewalks and crossings, and considering integrating with other modes. | [48,49,68,93,113]. |
| XX. Selective Ranges: Traffic lanes with traffic restrictions for some means of transport, aiming at the flow of traffic on the roads. | [27,50,71,78,94]. |
| XXI. Safety: Actions aimed at improving the security of the transport infrastructure to ensure the physical integrity of the users of the roads and means of transport, as well as preventive actions. | [2,9,19,20,30,39,44,46,47,49,52,56,61,83, 93,96,99–101,106,108,114,115,118,121, 123,126]. |
| XXII. Logistic solutions: Solutions to optimize the commodity logistics chain, traffic networks, freight, and goods movement. | [6,14,46,51,66,86,106,128,135,138]. |
| XXIII. Cooperation between stakeholders: Inclusion of society's segments and public and private bodies that may affect or are affected by mobility services' results in discussions on improving these services to ensure that the innovations implementation and mobility-related interventions meet your interests. | [22,26,32,38,40,41,45,50,52,69,73,82,86, 105,130,136]. |
| XXIV. Continuous renovation of transport infrastructure: Modernization of the current infrastructure and the implementation of new solutions to ensure the capacity to meet the travel demand. | [4,19,22,25,48,54,65,71,76,90,93,112, 114,118,122,134,139]. |
| XXV. Maintenance: Initiatives for the conservation of roads, vehicles, equipment, and systems used in the mobility of cities. | [53,54,93,104,107]. |
| XXVI. Traffic accident detection and support system: A system formed by hardware, software, procedures, and facilities for real-time detection of traffic accidents and assistance to the injured. | [4,55,56,64,140]. |

## 4.2. Survey Results

The value of Cronbach's Alpha (0.922) confirmed the questionnaire's reliability and the data obtained. Figure 4 presents the respondents' profiles according to the demographic data obtained through the questionnaire's first section.

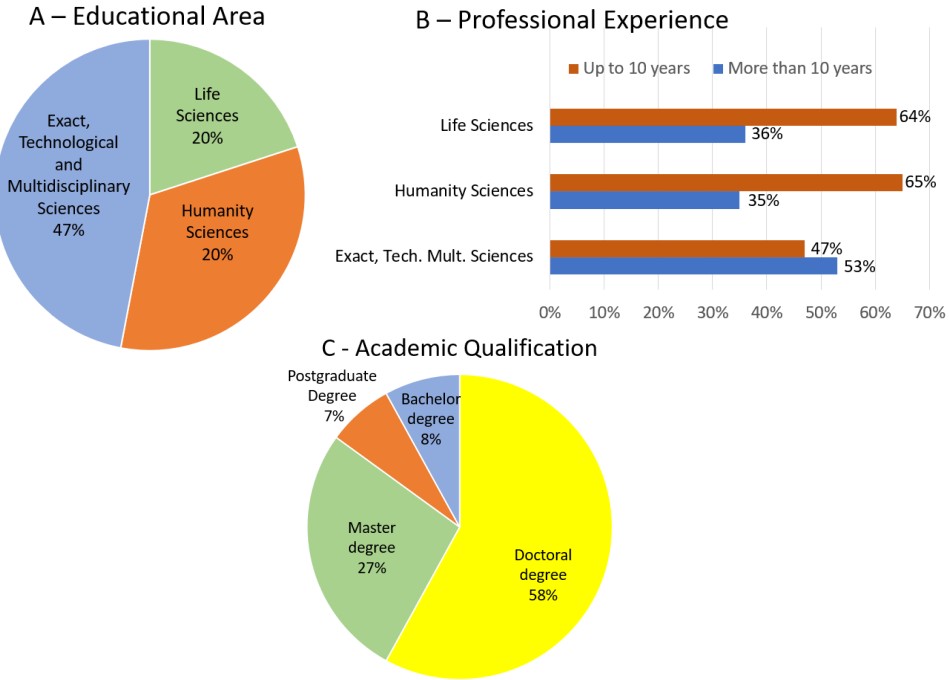

**Figure 4.** Demographic data; (**A**) Educational Area; (**B**) Professional experience; (**C**) Academic Qualifications.

Figure 5 shows the list of drivers prioritized by the relative median for the three Colleges of Knowledge and the entire sample. All drivers were considered important by the specialists (relative median greater than 3.0), corroborating the view of researchers who work with the theme. We consider drivers rated as "extremely important" (relative median equal to 5.0) as priorities for the College of Knowledge and the entire sample. Of the 12 drivers assessed as the most important by at least one of the Colleges of Knowledge, seven were considered a priority (Table 3).

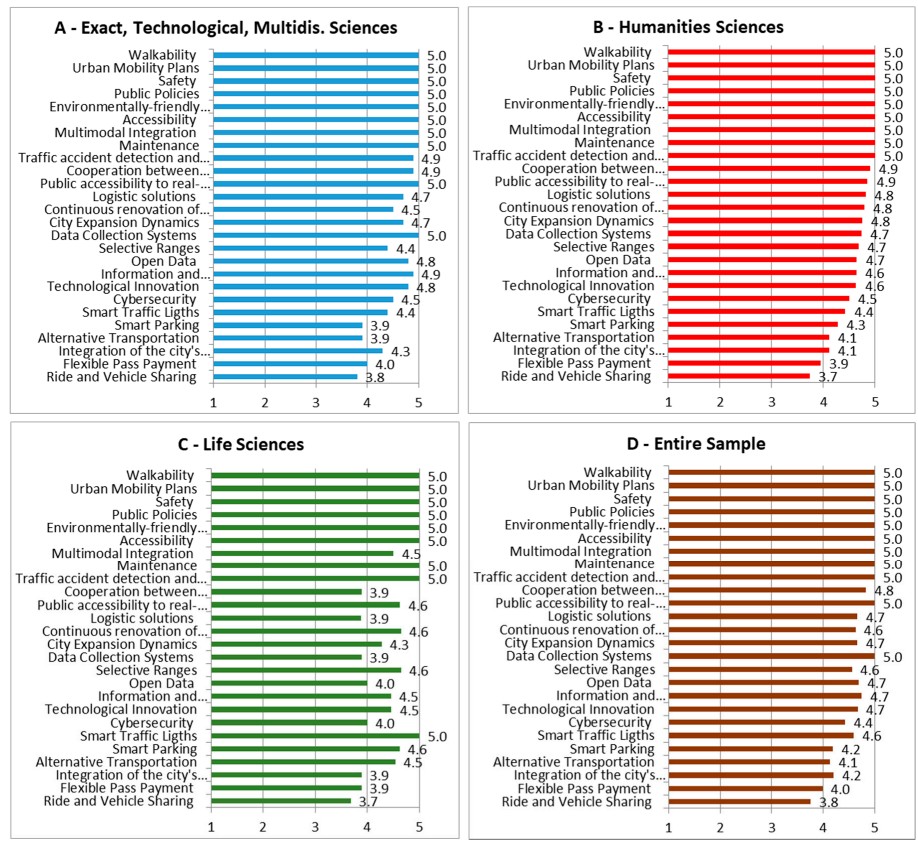

**Figure 5.** Drivers classified by the relative median for the three Colleges of Knowledge and total respondents; (**A**) Exact, Technological and Multidisciplinary Sciences; (**B**) Humanities Sciences; (**C**) Life Sciences; (**D**) Entire Sample.

**Table 3.** Drivers ranked as "extremely important".

| Guidelines | Humanities Sciences | Exact, Technological, Multidisciplinary Sciences | Life Sciences | Entire Sample |
|---|:---:|:---:|:---:|:---:|
| Urban Mobility Plans | 5 | 5 | 5 | 5 |
| Public Policies | 5 | 5 | 5 | 5 |
| Environmentally friendly policies | 5 | 5 | 5 | 5 |
| Accessibility | 5 | 5 | 5 | 5 |
| Walkability | 5 | 5 | 5 | 5 |
| Safety | 5 | 5 | 5 | 5 |
| Maintenance | 5 | 5 | 5 | 5 |
| Multimodal Integration | 5 | 5 | | 5 |
| Traffic accident detection and support system | 5 | | 5 | 5 |
| Data Collection Systems | | 5 | | |
| Public accessibility to real-time information | | 5 | | |
| Smart Traffic Lights | | | 5 | |

Figure 6 shows the drivers' relative behavior for the three College of Knowledges and the entire sample. It is possible to notice that the College of Knowledge of Exact, Technological and Multidisciplinary Sciences presents a different results pattern. The median found for most drivers that are not considered "extremely important" for mobility intelligence was lower than the other areas. Drivers such as Multimodal Integration, Cooperation Between Stakeholders, Logistic Solutions, Data Collection, Cybersecurity, and Information and Communication Technology received an evaluation well below that of the other College of Knowledges, in contrast to the evaluation trend observed in others.

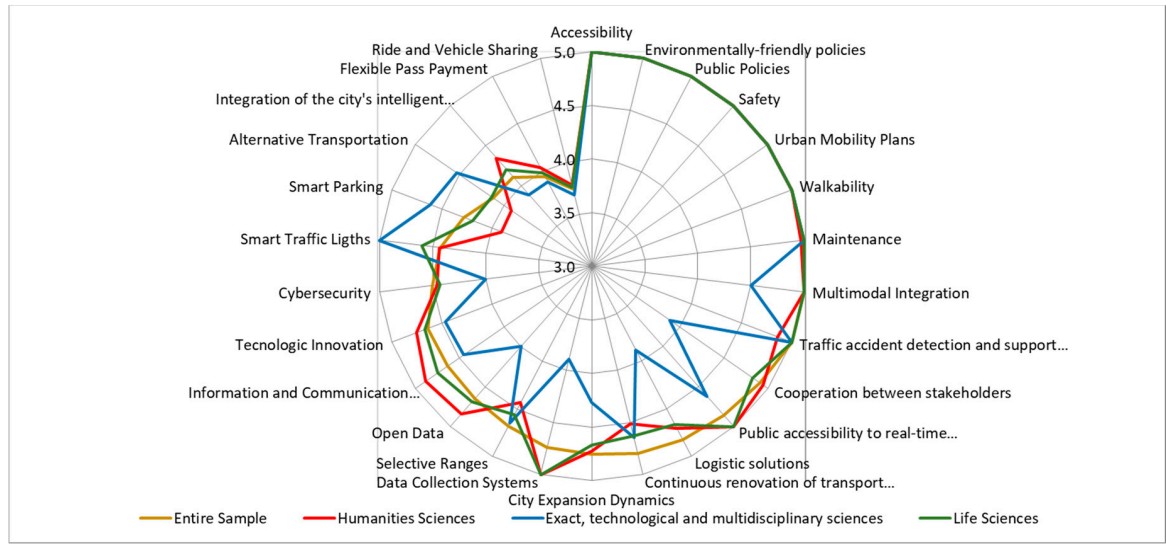

**Figure 6.** Drivers' behavior.

The 26 drivers identified can be grouped according to their main approach in three categories: city governance, technical solutions, and technological resources (Table 4):

**Table 4.** Drivers grouped by the approach.

| City Governance | Technical Solutions | Technological Resources |
|---|---|---|
| Urban Mobility Plans<br>Public Policies<br>Environmentally-friendly policies<br>Cooperation between stakeholders<br>Continuous renovation of transport infrastructure<br>Maintenance<br>City Expansion Dynamics<br>Safety | Multimodal Integration<br>Ride and Vehicle Sharing<br>Alternative Transportation<br>Accessibility.<br>Walkability<br>Selective Ranges<br>Logistic solutions | Information and Communication Technology<br>Data Collection Systems<br>Smart Traffic Lights<br>Smart Parking<br>Flexible Pass Payment<br>Cybersecurity<br>Technological Innovation<br>Public accessibility to real-time information<br>Open Data<br>Traffic accident detection and support system<br>Integration of the city's intelligent infrastructure with smart buildings |

Table 5 presents the drivers considered a priority in each approach.

Public policies are essential tools for directing government programs, actions, and decisions. They enable socially relevant and politically determined objectives to be achieved through the coordination of the means available to the state and private activities [68]. A set of goals, cohesive public policies, focusing on improving results and establishing the bases for action by public authorities and the private sector, drives the improvement of urban mobility, making investments in the sector more attractive, mainly by directing incentives and legal and financial guarantees.

**Table 5.** Priority drivers grouped by the approach.

| Governance | Technical Solutions | Technological Resources |
|---|---|---|
| Public Policies<br>Environmentally-friendly policies<br>Urban Mobility Plans<br>Maintenance<br>Safety | Accessibility<br>Walkability | - |

However, in Brazil and other developing countries, cities' accelerated growth has generated many problems along with few solutions. In many cases, governments have to adjust the solution of problems to existing public policies [91]. In this context, in a significant portion of Brazilian cities, public policies' main focus has not been to increase urban mobility intelligence but to solve structural problems. However, it is important to emphasize that the solution of structural problems is essential for improving the intelligence of the city's services [70], which means that even though the increase in intelligence is not the main focus, it happens as a consequence.

Among public policies, environmentally friendly policies have received special attention. Mobility solutions have been criticized for not being able to address aspects related to sustainability challenges effectively. It has been a challenge for the mobility sector to develop solutions that reduce the impact on sustainability [2,130].

Sustainability and smart cities are intrinsically related themes [92], and it is ineffective to address smart mobility without associating it with aspects of sustainability and vice versa. Also, the concept of intelligent mobility is intrinsically related to intelligent and sustainable transport, which has meant that a significant portion of the studies found in the literature and policies of many countries have focused on improving the sustainability and intelligence of transport. In Europe, for example, transport policy over time has focused on issues of transport sustainability [93].

In Brazil, the main public mobility policy is the National Urban Mobility Policy [141], which aims to (a) reduce inequalities and promote social inclusion; (b) promote access to basic services and social facilities; (c) provide improvement in the urban conditions of the population concerning accessibility and mobility; (d) promote sustainable development by mitigating the environmental and socioeconomic costs of displacing people and cargo in cities; and (e) consolidating democratic management as an instrument and guarantee for the continuous construction of the improvement of urban mobility.

In several countries, the instrument that enforces the urban mobility policy is the urban mobility plan, which in Brazil considers short, medium, and long-term scenarios, establishes goals, strategic actions, and resources to improve urban mobility. Although Brazilian legislation establishes that the policy is mandatory for municipalities with more than 20,000 inhabitants, the legal deadlines for meeting this requirement have been postponed due to the lack of structure and financial resources in several of them.

This whole context, coupled with the fact that Brazil has faced a serious political and financial crisis that has deteriorated basic services in the city and increased the perception by the society of political inefficiency, lack of planning, and the inability of public agencies with proposing solutions, may have influenced the experts' judgment of considering public policies, environmentally friendly policies and urban mobility plans drivers as priorities.

Concerning driver maintenance, the main focus is on conserving roads, vehicles, equipment, and systems used in cities' mobility. Intelligent mobility demands planning that considers public roads' periodic maintenance guided by information from digital monitoring systems. Moreover, public agencies and service concessionaires need to have open channels with citizens who use the transport network to receive notices and inspect the transport network.

Maintenance activities are usually expensive, which is an important barrier for many countries. In this context, actions aimed at increasing the intelligence of the maintenance system, which consider its

objectives, strategies, and processes in an integrated manner, in addition to improving the performance of the transport network, can be an important ally for cost reduction.

Intrinsically related to the maintenance driver, the safety driver has as its main focus actions aimed at improving the safety of the transport infrastructure to guarantee the physical integrity of the users on the roads and means of transport, as well as on prevention and control of criminal actions against pedestrians. The solutions usually used for smarter mobility, such as, for example, improving public transport, bike lanes, intelligent traffic lights, and cameras, alone contribute to increased safety; that is, improving safety is an intrinsic characteristic of smart solutions.

Regarding the technical solutions approach, two intrinsically related drivers were considered priorities: accessibility and walkability. The accessibility driver is primarily focused on improving access for people with disabilities to the transportation infrastructure. The urban space is used by many people with different characteristics and needs, among which a significant portion has some degree of disability or mobility. An intelligent transport infrastructure collaborates so that people can move around with equal opportunities, having access to the transport network through their own effort and able to access any point in the city.

The walkability driver has as its main focus the stimulation of walking as a means of locomotion from improving the fluidity of walking and safety conditions. The actions to stimulate walking should consider the aspects related to the level of security against criminal actions and the conditions of the sidewalks and crossings, especially concerning the state of the pavement, elimination of obstacles, and public lighting. Walkability becomes more efficient when the transport network offers simple and ecologically friendly options to complement citizen mobility, which implies integration with other modes.

The accessibility and walkability drivers are intrinsically related since the solutions powered by the first collaborate to improve the results of the second and vice versa.

With regards to the scope of the results, the set of 26 drivers was identified from broad and detailed bibliographic research in the main knowledge bases and the main scientific publishers' websites, which means that they can be considered for the increase of the mobility intelligence from other countries. Concerning the set of priority drivers, can they also be considered as priorities for increasing cities' mobility intelligence in other countries? Well, Each country presents characteristics that differentiate them. However, the Brazilian reality is similar to that of many countries. This means that all aspects of the Brazilian reality presented in this work are experienced by most cities in underdeveloped and developing countries to a greater or lesser extent.

In summary, the Brazilian reality presented throughout the discussion is that of deteriorating essential services in the city, which has increased society's perception of political inefficiency, lack of planning, and public agencies' inability to propose solutions. There is also a perception of a lack of adequate environmental protection policies.

Regarding the applicability of this study's results to developed countries, it is important to highlight that the urban systems of these countries have gone through stages of evolution that have not yet occurred in underdeveloped and developing countries, to a greater or lesser extent. We believe that in countries that have a more consolidated infrastructure, a technological bias should prevail.

## 5. Conclusions

Smart mobility has been a theme increasingly present in the sustainability agendas in response to the impacts of transportation systems in cities. It is considered a key factor for cities to become more intelligent, making it important to identify drivers to improve mobility intelligence.

Based on extensive and detailed bibliographic research, we identify 26 drivers that increase urban mobility intelligence, determined by 181 professionals working in the field. Of the seven drivers considered a priority, five are related to governance and two to technical solutions, which suggests that governance actions are the main problem faced by cities concerning mobility.

Considering the scarcity of public resources resulting from the financial crises faced by Brazil and most underdeveloped and developing countries, we consider it essential to prioritize these drivers to contribute so that city managers can direct their efforts to the most important ones.

Of the seven drivers considered a priority by professionals working in the concerned field, five are related to governance and two to technical solutions, which suggests that governance actions are the main problem faced by cities concerning mobility.

The survey results confirmed the authors' view of the consulted works since all drivers were assessed as important by the respondents. However, it is important to note that research based on an evaluation of specialists has some degree of subjectivity resulting from the evaluator's interpretation of what is being evaluated.

The most recommended is that all drivers considered important are the focus of the managers' actions. However, considering the scarcity of public resources resulting from Brazil's financial crises and by most underdeveloped and developing countries, we believe it essential to prioritize these drivers to contribute so that city managers can direct their efforts to the most important ones.

This work also presents the typical limitation of studies based on bibliographic research to support their analysis. Although we have carried out broad and detailed bibliographic research, there is always a risk that a relevant article has not been included.

The survey results showed that due to the drivers' relationship, they could be grouped into three approaches: governance, technical solutions, and technological resources. It also showed that of the twenty-six drivers found in the literature, seven could be considered a priority: public policies, environmentally friendly policies, urban mobility plans, maintenance, safety, accessibility, and walkability, and that technological resources approach drivers were not considered a priority.

Concerning policymakers, the results show the importance of public policies for increasing the system's intelligence. From the perspective of priority drivers, we expect policymakers to create an environment in which guidelines and rules improve the sector's sustainability and the ability to incorporate new technologies, which enable new solutions that improve the system's ability to meet expectations and stakeholder needs. In this context, the results also underscore the importance of policies related to accessibility and walkability. It is also essential to focus on improving the sector's attractiveness for investors since increasing the system's intelligence demands new investment.

Concerning managers, for participating in the system's management and operation, they have the information and knowledge necessary to improve policymakers' and investors' decision-making processes since the path to smarter mobility depends on diagnosing the current situation. Also, priority drivers suggest that managers should pay special attention to urban mobility plans, maintenance, safety, accessibility, and walkability.

To improve the results of this research, we have two suggestions. The first is a survey about the barriers that make it challenging to obtain the drivers' results, especially the priority drivers. The second is surveying experts of other countries to compare differences and similarities of perceptions.

**Author Contributions:** Conceptualization, survey, data curation, methodology, writing—original draft, formal analysis, writing—review and editing, P.A.M.S.A.M. and C.A.P.S.; visualization, writing—review and editing, F.d.C.D., C.K.C., A.L.A.G., J.A.N.d.S. and W.d.S.eS.; supervision, C.A.P.S. All authors have read and agreed to the published version of the manuscript.

**Funding:** This research received no external funding.

**Acknowledgments:** The authors would like to thank all the experts who answered the survey. The authors also express their gratitude to the editor and anonymous reviewers for comments and suggestions.

**Conflicts of Interest:** The authors declare no conflict of interest.

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
