# Peer review of "Smart Mobility: The Main Drivers for Increasing the Intelligence of Urban Mobility"

_sustainability, doi:10.3390/su122410675_

Round 1

Reviewer 1 Report

The paper addresses important theme of smart mobility and indentify importance of drivers for increasing the intelligence of urban mobility. The theme is highly relevant in the context of nowadays societies. The paper has a nice flow and is easy to read. 

Here are my comments:

In the introduction authors should also mention previous research in the field with citations. They nicely explain the problem from the relevance point of view, but for scientific paper, they should upgrade the content also by previous research. In the introduction authors should also present the research question, which is also the motivation for their research. 

In the Chapter Materials and Methods the authors should provide more information about the main scientific journals' that were addressed in their research (in the current version, there is no information about it). 

Definitions of selected drivers are presented in the results, although this step could be a part of research design or even the literature review, because in current version of the paper, literature review is related more towards generic themes and not to identification and explaination of the factors.

The results - importance of drivers are presented based on colleges of knowledge. It could be interesting to test, if there are any significant differences in opinion (importance of the factors) between different groups (colleges of knowledge). 

Discussion should also present discussion of your results with previous research conducted in your field. This is related also to literature review that you have done and presented in the paper. 

I hope you will find these comments useful for the improvement of your paper. 

Author Response

Response to Reviewer 1 Comments

Dear Reviewer,

Here we are presenting our explanations for the comments received, which we agree with all. We are also attaching a version of the paper with the highlighted changes since we have kept Word in change tracking mode.

We take this opportunity to thank you for the time and effort you have taken to review our paper. The comments received undoubtedly improved the quality and level of the paper's understanding.

We hope this new version is more suitable for publication.

Regards,

The authors

Point 1. In the introduction authors should also mention previous research in the field with citations. They nicely explain the problem from the relevance point of view, but for scientific paper, they should upgrade the content also by previous research. In the introduction authors should also present the research question, which is also the motivation for their research. 

Although the urban mobility concept is not new, we did not find in the researched literature articles that aim to identify drivers that collaborate to increase mobility intelligence considering this as a whole. What we found was some researches addressing issues related to smart mobility. These works were summarized in item 3.1. Bibliographic research.

We insert the following text to clarify the research questions:

We address two main issues. The first was: what are the main drivers for increasing urban mobility intelligence? To address this question, we performed a broad and detailed bibliographic research considering the recommendations the Preferred Reporting Items for Systematic Reviews and Meta-Analyzes (PRISMA). The second question was: what are the priority drivers for increasing urban mobility intelligence? To addresses this question, we surveyed 181 professionals working in the concerned field to confirm the driver's importance and prioritize them.

Point 2. In the Chapter Materials and Methods the authors should provide more information about the main scientific journals' that were addressed in their research (in the current version, there is no information about it). 

We insert the following text:

Although all selected articles contributed to substantiate this work, some contributed more. Below we will present a summary of each one, consolidating excerpts written by the authors.

Getting smart about urban mobility – Aligning the paradigms of smart and sustainable [7]: The paper puts forward and explores the definition of smart urban mobility to help draw the paradigms of smart and sustainable closer together towards a common framework for urban mobility development. This article seeks to get to the bottom of what we think we mean or should really mean when we label urban mobility ‘smart.’ Its writing has been motivated by a concern that we may be at risk of reserving smart for a focus on technology enablement. There is a need to ensure that the paradigms of smart urban mobility and sustainable urban mobility are aligned.

Smart Cities Concept: Smart Mobility Indicator [11]: The authors analyze and discuss the solutions employed as smart mobility solutions to test smart cities' effectiveness. As a result of the analysis, the author proposes an indicator to analyze which solutions are more intelligent within the possible ones to implement to improve urban mobility. The indicator is based on parameters in the following areas: transport infrastructure, information infrastructure, mobility methods and vehicles used, and legislation.

A comprehensive view of intelligent transport systems for urban smart mobility [14]: The authors conduct an in-depth analysis of intelligent transport systems' role in supporting urban mobility to identify the main knowledge gaps in the literature. Most of the articles analyzed were related to technology, and, according to the author, there was a lack of quantitative models of analysis.

Smart mobility transition a socio-technical analysis in the city of Curitiba [28]: The authors studied the case of Curitiba, located in the south of Brazil and recognized as a smart city, stands out for its pioneering project of the Bus Rapid Transit (BRT) system. They analyze urban mobility based on the socio-technical transition of innovation and the multi-level perspective. They also have developed a timeline of the initiatives considered as milestones towards the transition to smart mobility and proposed a reflection about socio-technical innovation approach applied to smart urban mobility.

Encouraging sustainable mobility behaviour by designing and implementing policies with citizen involvement [33]: The paper discusses the theoretical concepts, design considerations and preliminary findings from Smart Mobility. It also presents the results of a project that aims to devise measures to encourage the increased use of public and non-motorized transport, integrating the principles of behavioral economics in public policies. The article describes the energy policies behind the project and outlines the theoretical framework for integrating behavioral perceptions in technology design and public policy.

A taxonomy for planning and designing smart mobility services [62]: The authors studied state of the art in mobility services through publications and intelligent mobility services provided by cities worldwide. The author proposed a taxonomy to plan and design intelligent mobility services considering the following dimensions: type of services, maturity level, users, applied technologies, delivery channels, benefits, beneficiaries, and common functionality.

An Interdisciplinary Review of Smart Vehicular Traffic and Its Applications and Challenges [116]: The Authors discuss the main application areas of smart traffic and smart mobility, synthesizing different perspectives. Some research challenges pertinent to sustainability, insurance, simulation and the handling of ambiguous information we are also highlighted.

Point 3. Definitions of selected drivers are presented in the results, although this step could be a part of research design or even the literature review, because in current version of the paper, literature review is related more towards generic themes and not to identification and explanation of the factors.

We rewrote the literature review to address all drivers.

Point 4. The results - importance of drivers are presented based on colleges of knowledge. It could be interesting to test, if there are any significant differences in opinion (importance of the factors) between different groups (colleges of knowledge). 

We rewrote the text referring to Figure 5 that shows the drivers' relative behavior for the three Colleges of Knowledge and the entire sample.

Figure 5 shows the drivers' relative behavior for the three Colleges of Knowledge and the entire sample. It is possible to notice that the College of Exact, Technological, and Multidisciplinary Sciences presents a different results pattern. The median found for most drivers that are not considered "extremely important" for mobility intelligence was lower than the other areas. Drivers such as Multimodal Integration, Cooperation Between Stakeholders, Logistic Solutions, Data Collection, Cybersecurity, and Information and Communication Technology received an evaluation well below that of the other Colleges, in contrast to the evaluation trend observed in other Colleges.

Point 5. Discussion should also present discussion of your results with previous research conducted in your field. This is related also to literature review that you have done and presented in the paper. 

We agree that we made the mistake of not citing the articles that supported the discussions. We fixed this error

Reviewer 2 Report

I think the article is interesting although I think there is a need to improve several parts of the paper.

Firstly, the abstract must better describe the originality of the research and write something more innovative. I disagree that few studies focus on identifying drivers that enhance their intelligence, in urban mobility sector.

The end of the introduction must frame how the article will be developed, following the criterion of an international Journal such as Sustainability.

Sections 3 and 4 must explained better, explaining better the methodological evolution step by step, because that is not clear.
The conclusions must also better summarize the results of the research conducted, highlighting the future implications of the research and its originality. In a nutshell, the authors must also better describe the 26 drivers discovered.

Extensive editing of English language and style is strongly required.

Author Response

Response to Reviewer 2 Comments

Dear Reviewer,

Here we are presenting our explanations for the comments received, which we agree with all. We are also attaching a version of the paper with the highlighted changes since we have kept Word in change tracking mode.

We take this opportunity to thank you for the time and effort you have taken to review our paper. The comments received undoubtedly improved the quality and level of the paper's understanding.

We hope this new version is more suitable for publication.

Regards,

The authors

Point 1. Firstly, the abstract must better describe the originality of the research and write something more innovative. I disagree that few studies focus on identifying drivers that enhance their intelligence, in urban mobility sector.

We agree that the statement “few studies in the literature focus on identifying drivers that enhance their intelligence” can lead to inappropriate interpretations. We wanted to say that we did not find in the researched literature articles that aim to identify drivers that collaborate to increase mobility intelligence considering this as a whole. What we found were some articles presenting drivers that collaborate to increase the intelligence of some mobility services, such as, for example, xx service [], service [] and service []. Although a significant portion of the literature focuses on improving mobility services, we identified that variables considered in these studies have the potential to increase mobility intelligence. We decided to change the summary, introduction and conclusions to express these aspects better.

Abstract

Original text 

Urban mobility plays a fundamental role in the complex ecosystem of smart cities. Although urban mobility is considered a key factor for cities to become smarter, few studies in the literature focus on identifying drivers that enhance their intelligence. This study collaborates to fill this gap by researching the main drivers that increase urban mobility intelligence, based on extensive and detailed bibliographic research and the vision of professionals with experience in the concerned field. The results show that, out of the twenty-seven drivers identified in the literature, seven were considered a priority.

Changed text 

Urban mobility plays a key role in the complex smart cities ecosystem. It is considered a key factor for cities to become more intelligent, which makes it important to identify drivers to improve their intelligence. In this study, we research the main drivers with the potential to increase urban mobility intelligence and prioritize them. We conducted broad and detailed bibliographic research based on the recommendations of the Preferred Report Items for Systematic Reviews and Meta-analyzes (PRISMA) to improve the systematic review. We also surveyed 181 professionals working in the concerned field to confirm the driver's importance and prioritize them. The results show that the 27 drivers identified in the literature were considered important, of which seven related to city governance and technical solutions were considered the most important to increase the urban mobility intelligence.

Introduction

Original text 

The improvement of intelligence and the sustainability of urban mobility are intrinsically related to society's demands, represented by the different stakeholders, whose difference in expectations and needs increases the theme's complexity and makes it difficult to understand the factors that most influence the increase in their intelligence. Although the concept of urban mobility is not new, there are few studies on the main factors that must be considered to make cities more intelligent and sustainable. 

Changed text 

Although the urban mobility concept is not new, we did not find in the researched literature articles that aim to identify drivers that collaborate to increase mobility intelligence considering this as a whole. What we found was some researches addressing issues related to smart mobility. However, we identified factors that can collaborate to increase mobility intelligence in the discussions and results of several of these studies.

Conclusion

We made several changes to the conclusions.

Smart mobility has been an increasingly present theme in the sustainability agendas in response to the impacts caused by transportation systems in cities. It is considered a key factor for cities to become more intelligent, making it important to identify drivers to improve mobility intelligence.

Based on extensive and detailed bibliographic research, we identify 26 drivers that increase the urban mobility intelligence, determined by 181 professionals working in the concerned field. Of the seven drivers considered a priority, five are related to Governance and two to Technical solutions, which suggests that governance actions are the main problem faced by cities concerning mobility.

Considering the scarcity of public resources resulting from the financial crises faced by Brazil and most underdeveloped and developing countries, we consider it essential to prioritize these drivers to contribute so that city managers can direct their efforts to the most important ones.

The survey results confirmed the authors' view of the consulted works since all drivers were assessed as important by the respondents. However, it is important to note that research based on an evaluation of specialists has some degree of subjectivity resulting from the evaluator's interpretation of what is being evaluated.

The most recommended is that all drivers considered important are the focus of the managers' actions. However, considering the scarcity of public resources resulting from Brazil's financial crises and by most underdeveloped and developing countries, we believe it essential to prioritize these drivers to contribute so that city managers can direct their efforts to the most important ones.

This work also presents the typical limitation of studies based on bibliographic research to support their analysis. Although we have carried out broad and detailed bibliographic research, there is always a risk that a relevant article has not been included.

The survey results showed that due to the drivers' relationship, they could be grouped into three approaches: Governance, Technical solutions, and Technological resources. It also showed that of the twenty-six drivers found in the literature, seven could be considered a priority: Public Policies, Environmental-friendly policies, Urban Mobility Plans, Maintenance, Safety, Accessibility, and Walkability, and that Technological resources approach drivers were not considered a priority.

Concerning policymakers, the results show the importance of Public Policies for increasing the system's intelligence. From the perspective of priority drivers, we expect policymakers to create an environment in which guidelines and rules improve the sector's sustainability and the ability to incorporate new technologies, which enable new solutions that improve the system's ability to meet expectations and stakeholder needs. In this context, the results also underscore the importance of policies related to accessibility and walkability. It is also essential to focus on improving the sector's attractiveness for investors since increasing the system's intelligence demands new investments.

Concerning managers, for participating in the system's management and operation, they have the information and knowledge necessary to improve policymaker and investors' decision-making processes since the path to smarter mobility depends on diagnosing the current situation.  Also, priority drivers suggest that managers should pay special attention to urban mobility plans, maintenance, safety. accessibility, and walkability.

To improve the results of this research, we have two suggestions. The first is a survey about the barriers that make it challenging to get the drivers' results, especially the priority drivers. The second is surveying with experts of other countries to compare differences and similarities of perceptions. The second is surveying with experts from other countries to compare differences and similarities in perceptions.

Point 2. The end of the introduction must frame how the article will be developed, following the criterion of an international Journal such as Sustainability.

We insert the following text:

The paper is structured as follows: Section 2 presents the literature review, mainly addressing the drivers identified in the bibliographic research. Section 3 presents the procedures used to carry out bibliographic research, the identification of smart mobility drivers, the survey of expert opinions, the data analysis, and the structuring of the presentation of results and discussion. Section 4 presents and discusses the research results. Conclusions are given in Section 5

Point 3. Sections 3 and 4 must explained better, explaining better the methodological evolution step by step, because that is not clear.

We decided to rewrote the section 3. Materials and Methods to consider these aspects and better explain the other procedures.

Point 4. The conclusions must also better summarize the results of the research conducted, highlighting the future implications of the research and its originality. In a nutshell, the authors must also better describe the 26 drivers discovered.

We decided to rewrote the conclusions to consider these aspects and better explain the other procedures.

Point 5. Extensive editing of English language and style is strongly required.

We fixed some errors. A detailed review will be made at the end of the review process

Reviewer 3 Report

The Paper is focused on the actual topic – identification of the main drivers for the increase of urban mobility intelligence. I appreciate the logical structure of the paper, the complexity of the approach applied for the realization of presented study, and the exhausting literature review (141 sources). Important concepts are included (walkability, accessibility for citizens with special needs, security…).

Comments for improvement:

The abstract is too short. It should include more detailed information about the methodology applied to research the main drivers that increase urban mobility intelligence.

The authors mention that “few studies in the literature focus on identifying drivers that enhance their intelligence” Could you please underline which specific studies referenced in the literature review identified these drivers?

Please add to the introduction of manuscript information about the paper structure and a short description of its sections.

Is the proposed methodology for the identification of the main drivers for the incensement of urban mobility intelligence fully applicable in any environment? Will be the same drivers identified for different environments? That means not only for the Latin-American continent but also for Europe, Asia, etc. What aspects (e.g. cultural ones) should be taken into consideration for the successful implementation of the proposed methodology?

The authors mention an important aspect that is complicating objective and efficient realization of the research. They declare that “there is still no common understanding of precisely this concept between professionals and researchers”. How did you try to minimize the influence of this “obstacle” on your research?

Please provide some summarization/conclusion from the information presented in Table 1 - Definitions of Smart Mobility by researchers.

What will be the most complex definition of Smart mobility according to the knowledge from the literature review and experts’ evaluations?

How many professionals/experts were identified by applying the first approach described in section 3.3. “Survey of Expert Opinions”?

Was effectuated any pilot study to evaluate the appropriateness of the questionnaire designed for the collection of experts’ opinions? Please provide more detailed information about the questionnaire (number of questions, its structure, if any weights were used during the evaluation, etc.) and also the time data about this part of the research.

Section 3.2.” Identification of Smart Mobility drivers and challenges “ is not comprehensible enough. Please describe more extensively the procedure that was applied for the identification of drivers. “To be considered an important driver, the information or concept should be present in at least TWO scientific articles…” Why two? “The driver with the fewest citations has been described in three articles.” From these sentences, the reader can conclude that the most important criteria applied at this stage were the connection between papers via references. Is that true? Please explain it more clearly.

Author Response

Response to Reviewer 3 Comments

Dear Reviewer,

Here we are presenting our explanations for the comments received, which we agree with all. We are also attaching a version of the paper with the highlighted changes since we have kept Word in change tracking mode.

We take this opportunity to thank you for the time and effort you have taken to review our paper. The comments received undoubtedly improved the quality and level of the paper's understanding.

We hope this new version is more suitable for publication.

Regards,

The authors

Point 1. The abstract is too short. It should include more detailed information about the methodology applied to research the main drivers that increase urban mobility intelligence.

We agree. We rewrote the abstract to consider these aspects. 

Original text 

Urban mobility plays a fundamental role in the complex ecosystem of smart cities. Although urban mobility is considered a key factor for cities to become smarter, few studies in the literature focus on identifying drivers that enhance their intelligence. This study collaborates to fill this gap by researching the main drivers that increase urban mobility intelligence, based on extensive and detailed bibliographic research and the vision of professionals with experience in the concerned field. The results show that, out of the twenty-seven drivers identified in the literature, seven were considered a priority.

Changed text 

Urban mobility plays a key role in the complex smart cities ecosystem. It is considered a key factor for cities to become more intelligent, which makes it important to identify drivers to improve their intelligence. In this study, we research the main drivers with the potential to increase urban mobility intelligence and prioritize them. We conducted broad and detailed bibliographic research based on the recommendations of the Preferred Report Items for Systematic Reviews and Meta-analyzes (PRISMA) to improve the systematic review. We also surveyed 181 professionals working in the concerned field to confirm the driver's importance and prioritize them. The results show that the 27 drivers identified in the literature were considered important, of which seven related to city governance and technical solutions were considered the most important to increase the urban mobility intelligence.

Point 2. The authors mention that “few studies in the literature focus on identifying drivers that enhance their intelligence” Could you please underline which specific studies referenced in the literature review identified these drivers?

We agree that the statement “few studies in the literature focus on identifying drivers that enhance their intelligence” can lead to inappropriate interpretations. We wanted to say that we did not find in the researched literature articles that aim to identify drivers that collaborate to increase mobility intelligence considering this as a whole. What we found were some articles presenting drivers that collaborate to increase the intelligence of some mobility services. Although a significant portion of the literature focuses on improving mobility services, we identified that variables considered in these studies have the potential to increase mobility intelligence. We decided to change the abstract, introduction and conclusions to express these aspects better.

Introduction

Original text 

The improvement of intelligence and the sustainability of urban mobility are intrinsically related to society's demands, represented by the different stakeholders, whose difference in expectations and needs increases the theme's complexity and makes it difficult to understand the factors that most influence the increase in their intelligence. Although the concept of urban mobility is not new, there are few studies on the main factors that must be considered to make cities more intelligent and sustainable. 

Changed text 

Although the urban mobility concept is not new, we did not find in the researched literature articles that aim to identify drivers that collaborate to increase mobility intelligence considering this as a whole. What we found was some researches addressing issues related to smart mobility. Although the literature focuses mainly on improving mobility services, we identified factors that can collaborate to increase mobility intelligence in the discussions and results of several of these studies.

Conclusion

We made several changes to the conclusions.

Smart mobility has been an increasingly present theme in the sustainability agendas in response to the impacts caused by transportation systems in cities. It is considered a key factor for cities to become more intelligent, making it important to identify drivers to improve mobility intelligence.

Based on extensive and detailed bibliographic research, we identify 26 drivers that increase the urban mobility intelligence, determined by 181 professionals working in the concerned field. Of the seven drivers considered a priority, five are related to Governance and two to Technical solutions, which suggests that governance actions are the main problem faced by cities concerning mobility.

Considering the scarcity of public resources resulting from the financial crises faced by Brazil and most underdeveloped and developing countries, we consider it essential to prioritize these drivers to contribute so that city managers can direct their efforts to the most important ones.

The survey results confirmed the authors' view of the consulted works since all drivers were assessed as important by the respondents. However, it is important to note that research based on an evaluation of specialists has some degree of subjectivity resulting from the evaluator's interpretation of what is being evaluated.

The most recommended is that all drivers considered important are the focus of the managers' actions. However, considering the scarcity of public resources resulting from Brazil's financial crises and by most underdeveloped and developing countries, we believe it essential to prioritize these drivers to contribute so that city managers can direct their efforts to the most important ones.

This work also presents the typical limitation of studies based on bibliographic research to support their analysis. Although we have carried out broad and detailed bibliographic research, there is always a risk that a relevant article has not been included.

The survey results showed that due to the drivers' relationship, they could be grouped into three approaches: Governance, Technical solutions, and Technological resources. It also showed that of the twenty-six drivers found in the literature, seven could be considered a priority: Public Policies, Environmental-friendly policies, Urban Mobility Plans, Maintenance, Safety, Accessibility, and Walkability, and that Technological resources approach drivers were not considered a priority.

Concerning policymakers, the results show the importance of Public Policies for increasing the system's intelligence. From the perspective of priority drivers, we expect policymakers to create an environment in which guidelines and rules improve the sector's sustainability and the ability to incorporate new technologies, which enable new solutions that improve the system's ability to meet expectations and stakeholder needs. In this context, the results also underscore the importance of policies related to accessibility and walkability. It is also essential to focus on improving the sector's attractiveness for investors since increasing the system's intelligence demands new investments.

Concerning managers, for participating in the system's management and operation, they have the information and knowledge necessary to improve policymaker and investors' decision-making processes since the path to smarter mobility depends on diagnosing the current situation.  Also, priority drivers suggest that managers should pay special attention to urban mobility plans, maintenance, safety. accessibility, and walkability.

To improve the results of this research, we have two suggestions. The first is a survey about the barriers that make it challenging to get the drivers' results, especially the priority drivers. The second is surveying with experts of other countries to compare differences and similarities of perceptions. The second is surveying with experts from other countries to compare differences and similarities in perceptions.

Point 3. Please add to the introduction of manuscript information about the paper structure and a short description of its sections.

We insert the following text:

The paper is structured as follows: Section 2 presents the literature review, mainly addressing the drivers identified in the bibliographic research. Section 3 presents the procedures used to carry out Bibliographic research, the Identification of Smart Mobility drivers, the Survey of Expert Opinions, and Data analysis. Section 3 presents and discusses the research results. Conclusions are given in Section 5

Point 4. Is the proposed methodology for the identification of the main drivers for the incensement of urban mobility intelligence fully applicable in any environment? Will be the same drivers identified for different environments? That means not only for the Latin-American continent but also for Europe, Asia, etc. What aspects (e.g. cultural ones) should be taken into consideration for the successful implementation of the proposed methodology?

We decided to change the Materials and Methods, Introduction, and Results and Discussions to express these aspects better.

Materials and Methods

We structured the research question better and moved it to the Introduction

Original text 

This research's main question was: what are the most important drivers for increasing urban mobility intelligence? To answer this question, we have established a three-step approach: bibliographic research, field research with specialists in the field of mobility, and data analysis.

Changed text 

We use an approach widely used in research that aims to identify variables related to a given phenomenon and the importance degree of these variables. It consists of five steps: Bibliographic research, Identification of smart mobility drivers, Survey of expert opinions, and Data analysis, and Structuring the presentation of results and discussion.

In the Introduction we made the following change:

Original text 

The improvement of intelligence and the sustainability of urban mobility are intrinsically related to society's demands, represented by the different stakeholders, whose difference in expectations and needs increases the theme's complexity and makes it difficult to understand the factors that most influence the increase in their intelligence. Although the concept of urban mobility is not new, there are few studies on the main factors that must be considered to make cities more intelligent and sustainable. 

To collaborate to fill this gap, in this work, we identified in the existing literature the factors with the potential to increase the intelligence of urban mobility, and we prioritize these factors from the evaluation of experts on the subject.

Changed text 

The improvement of intelligence and the sustainability of urban mobility are intrinsically related to society's demands, represented by the different stakeholders, whose difference in expectations and needs increases the theme's complexity and makes it difficult to understand the factors that most influence the increase in their intelligence. Although the concept of urban mobility is not new, there are few studies on the main factors that must be considered to make cities more intelligent and sustainable. 

In this context, we address two main issues. The first was: what are the main drivers for increasing urban mobility intelligence? To address this question, we performed a broad and detailed bibliographic research considering the recommendations the Preferred Reporting Items for Systematic Reviews and Meta-Analyzes (PRISMA). The second question was: what are the priority drivers for increasing urban mobility intelligence? To addresses this question, we surveyed 181 professionals working in the concerned field to confirm the driver's importance and prioritize them.

Results and Discussions

Original text 

About the scope of the results, the set of twenty-six drivers was identified from a wide and detailed bibliographic search in the main knowledge bases and the main scientific publishers' websites, which means that they can be considered for the increase of the mobility intelligence from other countries. Concerning the set of drivers considered priorities, can they also be regarded as priorities for increasing cities' mobility intelligence in other countries? Each country presents a political, social, economic, and geographical reality, whose characteristics differentiate them. However, the Brazilian reality is similar to that of many countries, which means that all aspects of the Brazilian reality presented in this work, to a greater or lesser extent, are experienced by most cities in developing countries.

Changed text 

About the scope of the results, the set of twenty-six drivers was identified from a broad and detailed bibliographic research in the main knowledge bases and the main scientific publishers' websites, which means that they can be considered for the increase of the mobility intelligence from other countries. Concerning the set of priority drivers, can they also be considered as priority for increasing cities' mobility intelligence in other countries? Each country presents characteristics that differentiate them. However, the Brazilian reality is similar to that of many countries. It means that all aspects of the Brazilian reality presented in this work are experienced by most cities in developing and underdeveloping countries to a greater or lesser extent.

In summary, the Brazilian reality presented throughout the discussion is that of deteriorating essential services in the city, which has increased society's perception of political inefficiency, lack of planning, and public agencies' inability to propose solutions. There is also a perception of a lack of adequate environmental protection policies.

Regarding the applicability of this study's results to developed countries, it is important to highlight that the urban systems of these countries have gone through stages of evolution that have not yet occurred in underdeveloped and developing countries, to a greater or lesser extent.

Point 5. The authors mention an important aspect that is complicating objective and efficient realization of the research. They declare that “there is still no common understanding of precisely this concept between professionals and researchers”. How did you try to minimize the influence of this “obstacle” on your research?

We found that the statement “there is still no common understanding of precisely this concept between professionals and researchers” can lead to inappropriate interpretations. We did not want to affirm that the understanding of researchers and professionals is different. We decided to change this statement and include the concept of smart that we consider most appropriate.

Original text 

There is an increase in the use of "smart city" terminology in academia; however, there is still no common understanding of precisely this concept between professionals and researchers [9,10]. The concept of "smart city" has great potential to address several adverse effects of rapid urbanization [3] because it is linked to the implementation of several beneficial changes in the functioning of city dynamics [11].

Changed text 

There is an increase in the use of "smart city" terminology; however, there is no consensus on the smart city concept [8,9,10]. We agree with Guedes et al. {8} that "a more current and comprehensive way of understanding a smart city from the integration of existing knowledge and experiences is that of an innovative city, which combines aspects of intelligence and sustainability through a governance that integrates stakeholder interactions and uses the technology." The concept of "smart city" has great potential to address several adverse effects of rapid urbanization [3] because it is linked to the implementation of several beneficial changes in the functioning of city dynamics [11].

Point 6. Please provide some summarization/conclusion from the information presented in Table 1 - Definitions of Smart Mobility by researchers.

and

Point 7. What will be the most complex definition of Smart mobility according to the knowledge from the literature review and experts’ evaluations?

We insert the following text:

From these concepts, we can understand smart mobility as being mobility that uses digital technologies to integrate systems and means of transportation and interact with users, aiming at a sustainable, safe, accessible environment that meets citizens' mobility needs.

Point 8. How many professionals/experts were identified by applying the first approach described in section 3.3. “Survey of Expert Opinions”?

and

Point 9. Was effectuated any pilot study to evaluate the appropriateness of the questionnaire designed for the collection of experts’ opinions? Please provide more detailed information about the questionnaire (number of questions, its structure, if any weights were used during the evaluation, etc.) and also the time data about this part of the research.

and

Point 10. Section 3.2.” Identification of Smart Mobility drivers and challenges “ is not comprehensible enough. Please describe more extensively the procedure that was applied for the identification of drivers. “To be considered an important driver, the information or concept should be present in at least TWO scientific articles…” Why two? “The driver with the fewest citations has been described in three articles.” From these sentences, the reader can conclude that the most important criteria applied at this stage were the connection between papers via references. Is that true? Please explain it more clearly.

We decided to rewrote the section 3. Materials and Methods to consider these aspects and better explain the other procedures.

Round 2

Reviewer 1 Report

I agree with the changes made by the authors.

Reviewer 2 Report

the authors did all required revisions.

Reviewer 3 Report

All reviewer´s comments had been taken into consideration by the authors and in some way had been included in the new version of the paper. This led to better comprehensibility of the text and improved the quality of the presentation. A significant part of the text has been rewritten and the new version of the paper provides a more clear explanation of the methods and procedures applied. Applied format of the numbering of references is not in accordance with the Electronics Microsoft Word template file.